# PARTIAL SIMULATION FOR IMITATION LEARNING

## ABSTRACT

Model-based imitation learning methods require full knowledge of the transition kernel for policy evaluation. In this work, we introduce the Expert Induced Markov Decision Process (*e*MDP) model as a formulation of solving imitation problems using Reinforcement Learning (RL), when only partial knowledge about the transition kernel is available. The idea of *e*MDP is to replace the unknown transition kernel with a synthetic kernel that: a) simulate the transition of state components for which the transition kernel is known ($s_r$), and b) extract from demonstrations the state components for which the kernel is unknown ($s_u$). The next state is then stitched from the two components: $s = \{s_r, s_u\}$. We describe in detail the recipe for building an *e*MDP and analyze the errors caused by its synthetic kernel. Our experiments include imitation tasks in multiplayer games, where the agent has to imitate one expert in the presence of other experts for whom we cannot provide a transition model. We show that combining a policy gradient algorithm with our model achieves superior performance compared to the simulation-free alternative.

## 1 INTRODUCTION

Recent work on Imitation Learning (IL) offers a new approach to the problem. Torabi et al. (2018) and Baram & Mannor (2018) suggest defining success if the agent and the expert influence the environment in the same way and not if they necessarily take the same actions. They remove the need to label expert actions and allow to settle for examples that include nothing but state transitions. The objective function they define seeks to minimize the distance between the state-transition density functions induced by the agent and the expert. However, since the state-transition dynamics is usually complex and unknown, it can not be calculated explicitly and is instead estimated through sampling. For reasons such as cost, time and safety, the sampling occurs in simulation and not in the real plant. However, despite considerable advances in simulation technologies, it is still a limited tool when it comes to complex real-world problems. Therefore, the adoption of advanced IL methods is hindered.

Consider the self-driving car example. Given their intent, physical simulation of pedestrians and vehicles can be done with high accuracy. However, simulating the intent itself, i.e., the decision-making process of those entities, is challenging. In this example, simulating the transition of other road users is hard (requires intent and physical transformation), however, simulating the ego car that is controlled externally is feasible (requires physical transformation only). The unwelcome solution, in this case, is to resort to behavior cloning (BC) that do not require simulation at all (Pomerleau, 1991). However, as previously mentioned, BC follows a different success criterion that requires access to expert actions and is less in line with the true definition of success. But most importantly, BC discards two key ingredients of the problem: a) states order (i.e., it breaks trajectories), and b) partial knowledge about the transition kernel (ego car in the example above). In this paper we ask the following question:

> *Can we enjoy the benefits of contemporary imitation methods if we keep every trajectory intact, and maintain partial information on the transition kernel?*

We answer this question affirmatively and introduce a new model called *expert-induced* Markov Decision Process (*e*MDP) that fulfill this wish. *e*MDP transforms a given set of demonstrations into

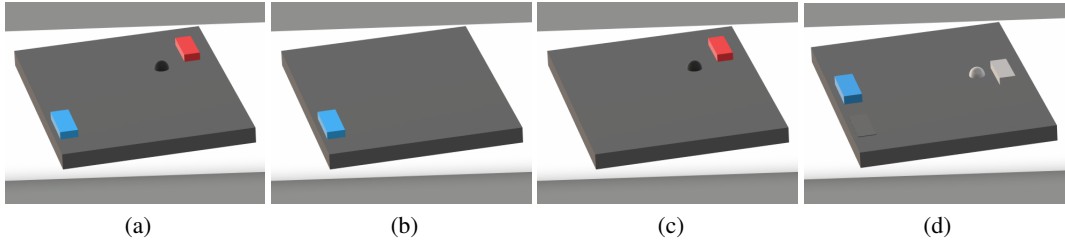

|  (a)  |  (b)  |  (c)  |  (d)  |

Figure 1: **Illustration of the transition kernel in an *e*MDP**. (a) two players are playing the game of Pong. We wish to imitate the left (blue) one at the presence of the right (red) expert. The state is split into two components: (b) *Responsive* components that we can simulate, and (c) *Unresponsive* components that we do not know how to simulate. (d) the *e*MDP model produces the next state by updating each component separately and then stitching them together. The unresponsive elements (gray) are extracted *as is* from the consecutive state in the demonstration, while the responsive element (blue player) is updated according to an external command. Notice that the original expert is faintly visible next to the blue agent for visualisation purposes.

a Markov Decision Process (MDP), and we show that solving it amounts to solving an imitation problem that seeks to match the state densities at each step. The idea of *e*MDP is to replace the unknown transition kernel with a synthetic kernel that: 1) simulate the transition of state components for which the transition kernel is known, and 2) extract from demonstrations the state components for which the kernel is unknown (see illustration in Figure 1). To understand the conditions when the use of *e*MDP is just, we derive a PAC result that bounds the error in the state-value function between the *e*MDP model and the genuine model that uses the original transition kernel. Finally, we show empirical results that stress the benefits of using *e*MDP when the transition kernel is partially available and model-based approaches are not applicable.

## 2 PRELIMINARIES

The following describes the mathematical formulation of MDPs and the assumptions we require to build an *e*MDP model. Following after, we outline the optimization problem solved by *e*MDP.

### 2.1 MATHEMATICAL FORMULATION

We assume an MDP defined as $\mathcal{M} = \{\mathcal{S}, \mathcal{A}, \mathcal{P}, r, \gamma\}$, where $\mathcal{S}$ is a continuous state space endowed with a metric $d_{\mathcal{S}}$, $\mathcal{A}$ is the action space, $\mathcal{P} : \mathcal{S} \times \mathcal{A} \times \mathcal{S} \to \mathbb{R}_+$ is the transition probability induced by a transition kernel $\mathcal{F} : \mathcal{S} \times \mathcal{A} \to \mathcal{S}$, $r : \mathcal{S} \times \mathcal{A} \to \mathbb{R}$ is the reward function and $\gamma \in [0, 1]$ is the discount factor. We also wish to define $\rho_0 : \mathcal{S} \to \mathbb{R}_+$ as the initial state distribution.

Our interest is in special MDPs where $\mathcal{S}$ can be factored as $\mathcal{S} = \mathcal{S}_r \times \mathcal{S}_u$, where $\mathcal{S}_r$ refers to state components that are governed by a *Responsive* kernel $\mathcal{F}_r : \mathcal{S} \times \mathcal{A} \to \mathcal{S}_r$. Put in words, $\mathcal{S}_r$ represents element in the state space that we can simulate (ego car in the example above). On the other hand, $\mathcal{S}_u$ represents the complement part of $\mathcal{S}$, governed by an unknown kernel $\mathcal{F}_u : \mathcal{S} \times \mathcal{A} \to \mathcal{S}_u$ (other road users in the example above). Therefore, it will be replaced by a synthetic kernel $\tilde{\mathcal{F}}_u : T \to \mathcal{S}$ that will be explained in Section 4. We also assume the existence of an expert policy $\pi_e : \mathcal{S} \times \mathcal{A} \to [0, 1]$ that is used to generate a set of state-only trajectories $\mathcal{D} = \left(s_e^{i,0}, ..., s_e^{i,T}\right)_{i=1}^n$.

To measure the distance between probability distributions we will use the Integral Probability Measure (IPM) formulation:

$$\mathcal{K}(p, q) = \sup_{\mathcal{G}} \left\{ \left| \int_{\mathcal{S}} g(s)p(s)ds - \int_{\mathcal{S}} g(s)q(s)ds \right| : g \in \mathcal{G} \right\} \tag{1}$$

The IPM formulation is attractive since it represents multiple distance measures that can be recovered under different choices of $\mathcal{G}$. For example, choosing $\mathcal{G} = \{g : ||g||_L \leq 1\}$, where $L$ is the Lipschitz constant of $g$ (see Definition 2.1), reduces Eq. (1) to the popular Wasserstein metric. Setting $\mathcal{G} = \{g : ||g||_{\mathcal{H}} \leq 1\}$, where $\mathcal{H}$ represents a Reproducing Kernel Hilbert Space (RKHS)[1], and

---

[1]Informally, a Hilbert space is a RKHS if the evaluation functional $L_s : g \to g(s)$ is continuous at any $g$.

Eq. (1) becomes the famous kernel-distance. Although the analysis in Section 5 applies to a set of IPM measures, hereinafter we will set $\mathcal{G} = \{g : ||g||_L \leq 1\}$ to invoke the Wasserstein metric, which was previously shown to be weak (Arjovsky et al., 2017).

**Definition 2.1** (Lipschitz continuity). *Define two metric spaces $(M_1, d_1), (M_2, d_2)$. Let $g : M_1 \rightarrow M_2$ be a function from $M_1$ to $M_2$. The Lipschitz constant of $g$ is given by:*

$$||g||_L = \sup \big\{ \frac{d_2(g(s_1), g(s_2))}{d_1(s_1, s_2)} : s_1, s_2 \in M_1 \big\}.$$

## 2.2 PROBLEM FORMULATION

Let $d_\pi^t, d_E^t$ denote the state distribution at time $t$, induced by the agent and the expert respectively. We wish to solve the following optimization problem:

$$\arg\min_\pi \sum_{t=1}^{\infty} \gamma^t \mathcal{K}(d_\pi^t, d_E^t). \tag{2}$$

However, we cannot calculate either of the distributions. While $d_E^t$ can at least be estimated based on $\mathcal{D}$, $d_\pi^t$ cannot even be estimated because $\mathcal{F}_r \subsetneq \mathcal{F}$. To solve this problem, we turn to use $\tilde{d}_\pi^t$, the distribution that is induced by $\tilde{\mathcal{F}} = \{\tilde{\mathcal{F}}_u, \mathcal{F}_r\}$ and which can be estimated in the simulation. To justify the replacement of $d_\pi^t$ by $\tilde{d}_\pi^t$ we propose the following proposition:

**Proposition 2.1** (*e*MDP Optimal Solution). *Let $\mathcal{K}(p, q)$ be an IPM measure over probability distributions $p, q$. Let $d_\pi^t, \tilde{d}_\pi^t$ denote the $t$-step state distributions induced by $\mathcal{F}$ and $\tilde{\mathcal{F}}$ respectively. Then, if $\sum_{t=1}^{\infty} \gamma^t \mathcal{K}(d_\pi^t, d_E^t) = 0$ then $\tilde{d}_\pi^t = d_\pi^t$ for all $t \geq 0$.*

*Proof.* See Appendix A. □

Proposition 2.1 ensures that the optimal solution of *e*MDP coincides with the optimal solution of problem (2). However, it does not provide any guarantees when the optimum is not reached. In Section 5 we bound the errors between the *e*MDP and the original MDP for any $\pi$. Motivated to replace $d_\pi^t$ with $\tilde{d}_\pi^t$ in equation 2 we end up with a tractable optimization problem:

$$\arg\min_\pi \sum_{t=1}^{\infty} \gamma^t \mathcal{K}(\tilde{d}_\pi^t, d_E^t) = \arg\min_\pi \sum_{t=1}^{\infty} \gamma^t \sup_{g \in \mathcal{G}} \mathop{\mathbb{E}}_{\substack{s^t \sim d_\pi^t \\ s_e^t \sim d_E^t}} |g(s^t) - g(s_e^t)|. \tag{3}$$

Denoting $r_t = -|g(s^t) - g(s_e^t)|$, it is easier to see that equation 3 takes the form of adversarial imitation learning (Ho & Ermon, 2016), with $g$ playing the role of the discriminator. It is also interesting to see that if $d_\pi^t, d_E^t$ were to describe transition densities, then (3) would coincide with GAILfO's objective function. In the same way, if we had chosen $\mathcal{G} = \{g : ||g||_L \leq 1\}$, then (3) would have taken the form of Wasserstein GAN (Arjovsky et al., 2017).

However, solving (3) requires to address two optimization problems concurrently, which complicates the problem and leads to many instability issues associated with training GANs (Salimans et al., 2016). Fortunately, we note a key feature in our setup that allows simplifying problem (3). At each step, *e*MDP exposes two states: the synthetic state $\{s_r^t, s_u^t\} \sim \tilde{d}_\pi^t$ and the reference state $\{s_{r,e}^t, s_{u,e}^t\} \sim d_E^t$ that *share the same unresponsive component*: $s_u^t = s_{u,e}^t$. Therefore, we know that to achieve the optimal solution of (3) we must have that $s_r^t = s_{r,e}^t$. This condition can be enforced explicitly without invoking a neural-network based classification network. For example, by restricting $\mathcal{G}$ to be a class of norm function. If this is the case, then using the help of Lemma 2.1 we can upper bound (3) in the following way:

$$\arg\min_\pi \sum_{t=1}^{\infty} \gamma^t \sup_{g \in \mathcal{G}} \mathop{\mathbb{E}}_{\substack{s^t \sim d_\pi^t \\ s_e^t \sim d_E^t}} |g(s^t) - g(s_e^t)| \underset{g \text{ is a norm}}{\leq} \arg\min_\pi \sum_{t=1}^{\infty} \gamma^t \sup_{g \in \mathcal{G}} \mathop{\mathbb{E}}_{\substack{s^t \sim d_\pi^t \\ s_e^t \sim d_E^t}} |g(s^t - s_e^t)|. \tag{4}$$

**Lemma 2.1** (Norm Function Inequality). *Let $g$ be a norm function. Then $\forall x, y$ the following holds:*

$$|g(x) - g(y)| \leq |g(x - y)|$$

*Proof.* See Appendix B. □

To conclude, denoting $r(s_r^t, s_{r,e}^t) = -g(s_r^t - s_{r,e}^t)$, where $g(x) = ||x||_{d_S}$, we arrive at the final optimization problem of the $e$MDP model:

$$\arg\max_{\pi} \sum_{t=1}^{\infty} \gamma^t \mathop{\mathbb{E}}_{\substack{s^t \sim d_\pi^t \\ s_e^t \sim d_E^t}} r(s_r^t, s_{r,e}^t). \tag{5}$$

## 3 RELATED WORK

Solving problem (5) is challenging. From one hand, the constraint that $\mathcal{F}_r \subsetneq \mathcal{F}$ prohibits using model-based (i.e., simulation-based) methods. On the other hand, the constraint on inaccessible expert actions prohibits the use of BC or other simulation-free methods. This setup differs from previous imitation learning configurations that were introduced in the literature. As no method exactly fits problem (5), we turn to review the ones that can operate under slight modifications.

**Methods requiring access to $\mathcal{F}, \mathcal{A}, \mathcal{S}$:** A fairly permissive simulation-based setup that violates our setup twice. First, for assuming access to expert actions, and second, for assuming full access to $\mathcal{F}$. Algorithms of this type execute a simulation step $\zeta \sim \pi$ followed by a policy improvement step. DAGGer (Ross et al., 2011), an online no-regret algorithm, queries the expert at the states of $\zeta$ to generate ground-truth actions. Policy improvement is carried out by minimizing a loss function between expert and agent actions. GAIL (Ho & Ermon, 2016) and mGAIL (Baram et al., 2016) use $\zeta$ to train a discriminator to classify between fake and authentic trajectories. The discriminator's classification probability is used as a reward signal that the agent seeks to maximize.

**Methods requiring access to $\mathcal{A}, \mathcal{S}$:** Included in this group are simulation-free methods that assume access to expert actions. The most prominent approach of this class is BC that administers supervised learning tools to the problem. I.e., it tries to directly learn a state-to-action mapping. The strength of BC lies in its simplicity and is favorable when enough state-action pairs are available. When this is not true, BC will tend to suffer compounding errors. Extensions of BC include Bojarski et al. (2016) that learn a mapping from images to vehicle steering commands, andTai et al. (2016); Giusti et al. (2016) that learn a mapping between depth images to robot commands.

**Methods requiring access to $\mathcal{F}, \mathcal{S}$:** These are imitation algorithms that can train policies from state-only trajectories when full access to $\mathcal{F}$ is available. Torabi et al. (2018) follows GAIL's structure but uses a discrimination rule that is based on the state-transition distribution and not on the state-action joint distribution. Baram & Mannor (2018) describes a similar setup, however, there, the reward is modified to enhance stability using ideas of Preference-Based RL.

**Methods requiring access to $\mathcal{S}$:** The most stringent approach in terms of prior knowledge is also the only approach that can operate under the configuration proposed here. One notable example is Time Contrastive Networks (TCN) (Sermanet et al., 2018) that proved able to train a robot controller with zero-knowledge of $\mathcal{F}$. However, TCN also requires multiple views (cameras) of the agent to achieve good performance.

**Methods requiring access to a recovered model $\hat{\mathcal{M}}$:** Although not referred to explicitly, it is always possible, yet presumptuous, to use the demonstrations to learn a model $\hat{\mathcal{M}}$ of the environment, or at least parts of it. Depending on what MDP components are recovered, different approaches can be invoked as listed above. For example, a complete recovery of the underlying MDP, including the reward function $r$, also known as Inverse Reinforcement Learning (IRL) (Ng et al., 2000; Ziebart et al., 2008), allows administering any RL algorithm with no constraints what so ever.

## 4 THE $e$MDP MODEL

The motivation of $e$MDP is borrowed from techniques for learning time-series models from demonstrations (Venkatraman et al., 2015). Cascading prediction errors from forward-simulation with a learned model can result in predicting infeasible states. To solve this, the demonstrations are used to generate synthetic examples for the learner to ensure that prediction returns to typical states. In

the same spirit, we propose to construct an MDP that allows recovering from "off-route" states, even with partial knowledge of the state dynamics. $e$MDP provides guidance that compares the responsive elements in the agent's trajectory ($\mathcal{S}_r$) to the responsive elements in the original trajectory ($\mathcal{S}_{r,e}$). In the following, we outline the elements of the $e$MDP model.

**Action Space** $\mathcal{A}$**:** The action space of an $e$MDP can be defined arbitrarily and there is no requirement that $\mathcal{A} = \mathcal{A}_E$ since $e$MDP ignores expert actions.

**State Space** $\mathcal{S}$**:** We assume a decomposable state space $\mathcal{S} = \{\mathcal{S}_r, \mathcal{S}_u\}$. $\mathcal{S}_r$, where we demand that $\mathcal{S}_u \equiv \mathcal{S}_{u,e}$. There is also a similar demand on the responsive part $\mathcal{S}_r$, however this can be relaxed provided that a suitable reward function is available (see below).

**Reward Function** $r$**:** The reward is derived directly from (5), and is defined over the metric of $\mathcal{S}$: $r_t = -||s_r^t - s_{r,e}^t||_{d_\mathcal{S}}$. In Appendix C we demonstrate how additional shaping of $r_t$ is possible with the help of prior knowledge about the correct relations between $s_r$ and $s_{u,e}$.

**Discount Factor** $\gamma$**:** The discount factor lies in the same range of $[0,1)$. We denote $\gamma = 0$ as the BC-regime of $e$MDP (see Section 6.1), and $\gamma$ close to one as the RL-regime of the model. At the BC-regime, $e$MDP encourages $\pi$ to be greedy w.r.t the short-horizon-expert, while at the RL-regime it encourages $\pi$ to follow the "long-horizon-expert". Short-term experts can guide $\pi$ to fix immediate drifts that occur after short action sequences (easier). Long-term experts will help it to fix drifts that compound after long action sequences (harder). Therefore $e$MDP naturally supports $\gamma$-annealing.

**Transition Kernel** $\mathcal{F}$**:** Upon receiving an action, the new state is calculated in the following way:

- Simulating the transition of the responsive component: $s_r^{t+1} = \mathcal{F}_r(s^t, a^t)$.
- Reflecting the transition of the unresponsive component: $s_u^{t+1} \equiv \tilde{\mathcal{F}}_u(t) = s_{u,e}^{t+1}$.
- Stitching $s_r, s_u$ together to get the final state: $s^{t+1} = \{s_r^{t+1}, s_u^{t+1}\}$.

That is, the responsive component is updated according to $\mathcal{F}_r$, while the unresponsive component is extracted **as is** from the next state in the demonstration.

**Initial State Distribution** $\rho_0$**:** The initial state distribution is equivalent to the empirical state distribution of the demonstration set. I.e., initial states are sampled uniformly across time-steps and episodes in $\mathcal{D}$: $s_0 = \{s_r^0, s_u^0\} = \{s_{r,e}^{(i,t')}, s_{u,e}^{(i,t')}\}$, where $i \sim \mathbb{U}[1,...n]$ is the episode index and $t' \sim \mathbb{U}(1,...T)$ is the time index. For simplicity, hereinafter we omit the trajectory index $i$.

**Absorbing Set** $\mathcal{B}$**:** The termination set is defined by the set of states that the agent reaches after $T - t'$ steps. I.e., the last step in the current demonstration: $\mathcal{B} = \{s_t | t = T - t'\}$. In Appendix D we demonstrate how an augmentation of $\mathcal{B}$ is possible using prior knowledge.

## 5 THEORETICAL JUSTIFICATION OF THE $e$MDP MODEL

Lemma 2.1 ensures that $e$MDP perfectly follows the original MDP at the optimum solution. However, it tells us nothing otherwise. As long as the optimum is not met, $e$MDP induces an improper state-distribution which leads to an error. Denote the model governed by $\tilde{\mathcal{F}}$ as $\hat{\mathcal{M}}$, we are interested to compare the performance of the agent in $\hat{\mathcal{M}}$ to the one in the true model $\mathcal{M}$. The error we wish to calculate is the difference in the value function between the models, and in the following, we show how to derive it from samples of $\{s_r^t, s_{r,e}^t\}$. Due to space constraints, we deffer parts of the proof to the appendix and present here the highlights of the analysis.

**Analysis Overview** We rely on the IPM formulation introduced in Section 2 in the Wasserstein form. We desire to use the Wasserstein distance since it can metrize the weak topology of the state space. In this way, we can relate the error in a metric state space to a continuous distance in the distribution space. We will show how to estimate the distance from samples, without requiring access to the true, unavailable model $\mathcal{M}$. Since expert samples are scarce, it is also important to discuss the rate to which the empirical estimator converges to the real quantity, which we do so in Appendix H. With the empirical estimator at hand, and assuming that the transition kernel is a Lipschitz function (see Definition 5 in Asadi et al. (2018)), we are able to bound the divergence between the state distributions after a single step with high probability. Following right after, we

relate the 1-step divergence to the one that compounds after $n$ steps. All that is left is to use the $n$-steps divergence to bound the error in the value function.

**Empirical Estimation of the Single Step Divergence** Given two i.i.d sample sets: $\{S_{r,i}\}_{i=0}^m$, $\{S_{r,e,i}\}_{i=0}^m$ drawn randomly from $P_{\mathcal{M}}(\cdot|s,a)$ and $P_{\hat{\mathcal{M}}}(\cdot|s,a)$ respectively, the empirical estimator of $\mathcal{K}$ is given by:

$$\hat{\mathcal{K}}(\mathbb{P}_{\mathcal{M}}(\cdot|s,a), \mathbb{P}_{\hat{\mathcal{M}}}(\cdot|s,a)) = \sup_{f \in \mathcal{F}} \Big| \frac{1}{m} \sum_{i=0}^{2m} c_i f(\tilde{S}_i) \Big|, \tag{6}$$

where $\mathbb{P}_{\mathcal{G}}(\cdot|s,a) \doteq \frac{1}{m} \sum_{i=1}^m \delta_{\tilde{S}^{(i)}}$ represents the next-state empirical distribution under model $\mathcal{G}$, $c_i = +1$ when $\tilde{S}_i = S_r^{(i)}$ for $i = 1, ...m$ and $c_i = -1$ when $\tilde{S}_{m+i} = S_{r,e}^{(i)}$. This estimator is strongly consistent and converges to the population value as $m \to \infty$ as the following proposition suggests:

**Proposition 5.1.** *Let $(\mathcal{S}, d_{\mathcal{S}}$ be a totally bounded metric space. Then as $m \to \infty$:*

$$\Big| \mathcal{K}\big(P_{\mathcal{M}}(\cdot|s,a), P_{\hat{\mathcal{M}}}(\cdot|s,a)\big) - \hat{\mathcal{K}}\big(\mathbb{P}_{\mathcal{M}}(\cdot|s,a), \mathbb{P}_{\hat{\mathcal{M}}}(\cdot|s,a)\big) \Big| \xrightarrow{a.s} 0 \tag{7}$$

*Proof.* See Proposition 3.2 in Sriperumbudur et al. (2012). $\qquad\square$

The points we sample from the *e*MDP model are a function of a specific expert realization. Different realizations will lead to different estimations. In Appendix H we draw a PAC result on the convergence rate of the empirical estimator.

**From 1-step Divergence to $n-$step Divergence** In the following, we use the Lipschitz assumption on the transition function to bound the $n-$step error using the error that compounds after a single step[2].

**Lemma 5.1.** *Let $\mathcal{M}, \hat{\mathcal{M}}$ be two MDPs with Lipschitz transition functions with constants $L, \hat{L}$ respectively. Let $\bar{L} = \min\{L, \hat{L}\}$, and denote $\bar{K}_{\mathbb{P}_{\mathcal{M}, \hat{\mathcal{M}}}} = \max_{s \in \mathcal{D}} \hat{\mathcal{K}}\big(\mathbb{P}_{\mathcal{M}}(\cdot|s), \mathbb{P}_{\hat{\mathcal{M}}}(\cdot|s)\big) + \epsilon_{m,\delta,c,\nu,C_0,C_1}$, then for all $n \geq 1$ the following holds:*

$$P\Big( \mathcal{K}\big(P_{\mathcal{M}}^n(\cdot|\mu), P_{\hat{\mathcal{M}}}^n(\cdot|\mu)\big) \geq \bar{K}_{\mathbb{P}_{\mathcal{M}, \hat{\mathcal{M}}}} \sum_{i=0}^{n-1} \bar{L}^i \Big) \leq 2e^{-c}, \tag{8}$$

*where $P_{\mathcal{G}}^n(\cdot|\mu)$ is the distribution induced by model $\mathcal{G}$ after $n$ steps taken from distribution $\mu$.*

*Proof.* See Appendix K. $\qquad\square$

**Approximation Error in the State-Value Function** We are now ready to present the main result that relates the $n-$step divergence to the error in the value function. The following lemma is true for all $s$:

**Lemma 5.2.** *Let $\mathcal{M}, \hat{\mathcal{M}}$ be two MDPs with Lipschitz transition functions with constants $L, \hat{L}$ respectively such that $\bar{L} = \min\{L, \hat{L}\}$, and Lipschitz reward functions with a constant $L_R$. Then, $\forall s$, $\bar{L} \in [0, \frac{1}{\gamma}]$ and $n \geq 1$, with probability at least $1 - 2e^{-c}$ the following holds:*

$$|V_{\mathcal{M}}(s) - V_{\hat{\mathcal{M}}}(s)| \leq \frac{\gamma L_R \bar{K}_{\mathbb{P}_{\mathcal{M}, \hat{\mathcal{M}}}}}{(1-\gamma)(1-\gamma\bar{L})} \tag{9}$$

*Proof.* See Appendix I. $\qquad\square$

Not surprisingly, Lemma (5.2) show us that the correctness of the *e*MDP model depends on the Lipschitz of the reward function (which we control) and on the maximal error between the *e*MDP and the demonstration set: $\bar{K}_{\mathbb{P}_{\mathcal{M}, \hat{\mathcal{M}}}}$.

---

[2]Lemma 5.1 applies to several IPM measures besides the Wasserstein metric. The composition lemmas we prove in Appendix J can be used in the Kantarovich-Rubinstein duality theorem to generalize the result to compositions over transition function with bounded TV, Dudley or Wasserstein norms.

## 6 EMPIRICAL EVALUATION

The evaluation was made between the BC baseline and a Policy Gradient (PG) algorithm (Schulman et al., 2017) equipped with our $e$MDP model. Both agents used the same CNN-based policy (See Appendix L for details), and the same set of 10 demonstrations, each of a maximal length of 500 steps. In the following, we briefly describe the construction of the $e$MDP model for each experiment. Unless otherwise mentioned, both agents share the same action space.

**Pong-v0** a two-player tennis-like game (Brockman et al., 2016). We record 2 human players. Our goal was to imitate player1 at the presence of human player2. We do not have full knowledge about the dynamics of the problem because human player2 can not be simulated. Therefore, we resort to use our $e$MDP model in the following way: $\mathcal{F}_r = \{player1\}$, $\tilde{\mathcal{F}}_u = \{player2, ball\}$, and $\mathcal{A}$ consists of 5 vertical move commands in oppose to 3 movements used by the human experts.

**Surround** a two-player game where each player controls a growing "snake" comprised of the player's trajectory (Gam, 2019). The player who first touches either snakes loses. We let two human players to compete. The goal of the agent we trained was to imitate human player 1 at the presence of human player 2. The $e$MDP model was constructed in the following way: $\mathcal{F}_r = \{player1's\ snake\}$, $\tilde{\mathcal{F}}_u = \{player2's\ snake\}$ and the action space consists of two turning commands.

**Boxing** a two-player game that simulates a boxing match from a top view camera (Gam, 2019). We let two humans to compete. The goal of the agent we trained was to imitate human player 1 at the presence of human player 2. The $e$MDP was constructed in the following configuration: $\mathcal{F}_r = \{boxer1\}$, $\tilde{\mathcal{F}}_u = \{boxer2\}$, and $\mathcal{A}$ consists of 5 movement commands and two hit commands.

**Tennis** a two-player game simulating a tennis match (Brockman et al., 2016). We let two human players to compete. The goal of the agent we trained was to imitate human player 1 at the presence of human player 2. The following $e$MDP model was created: $\mathcal{F}_r = \{player1\}$, $\tilde{\mathcal{F}}_u = \{player2\}$ and the action space consists of 4 move commands and a single hit command.

**Assault-v0, Carnival-v0** a single-player shooter game (Brockman et al., 2016) where the player's character faces multiple enemy shooters. We trained an agent to imitate a human player while using a modified action space. Since we introduced modifications to the action space, we could no longer use the emulator of the game, and model-based imitation methods were impractical to use. We construct an $e$MDP model in the following way: $\mathcal{F}_r = \{player's\ shooter\}$, $\tilde{\mathcal{F}}_u = \{enemy\ shooters\}$ and the action space consists of seven movement commands and a single shoot command. The BC agent used the unmodified action space of the original game.

**Breakout-v0** a single-player game (Brockman et al., 2016) that do not require the construction of an $e$MDP model but is included here for its popularity. The $e$MDP was constructed in the following way: $\mathcal{F}_r = \{racket\}$, $\tilde{\mathcal{F}}_u = \{ball, bricks\}$ and the action space consists of seven horizontal move command. The BC agent used the unmodified action space of the human player.

The performance of the PG agent and BC are brought in Figure 2. The graphs present the cumulative return of the $e$MDP model as a function of the number of processed game frames. We note that this is an unfair comparison since the BC agent was trained to solve a different optimization problem, Therefore, we also provide video results for a visual comparison (see supplementary material).

### 6.1 QUALITATIVE EVALUATION

As shown in Figure 2, the PG agent clearly outperforms the BC baseline. In the following, we try to formalize the claim introduced in Section 1 that the solution of BC (to induce the same actions) deviates from the native objective of the problem (to induce the same effects on the environment). To do so, we compare the optimal solution of BC and $e$MDP. For a fair and clear comparison, we conduct the comparison at the BC Regime of $e$MDP ($\gamma = 0$) to encourage greedy policies that take into account a single-step horizon, much like the BC solution that completely overlooks transitions. Lemma 6.1 shows that the optimal policy assigns action probabilities proportionally to $r$:

**Lemma 6.1.** *The solution of optimization problem 5 with entropy regularization at the BC-regime ($\gamma = 0$) is given by:* $\pi^{RL}(a|s) = \frac{e^{r(s,a)/\tau}}{\sum_{\tilde{a}} e^{r(s,\tilde{a})/\tau}}$ .

*Proof.* See Appendix E. □

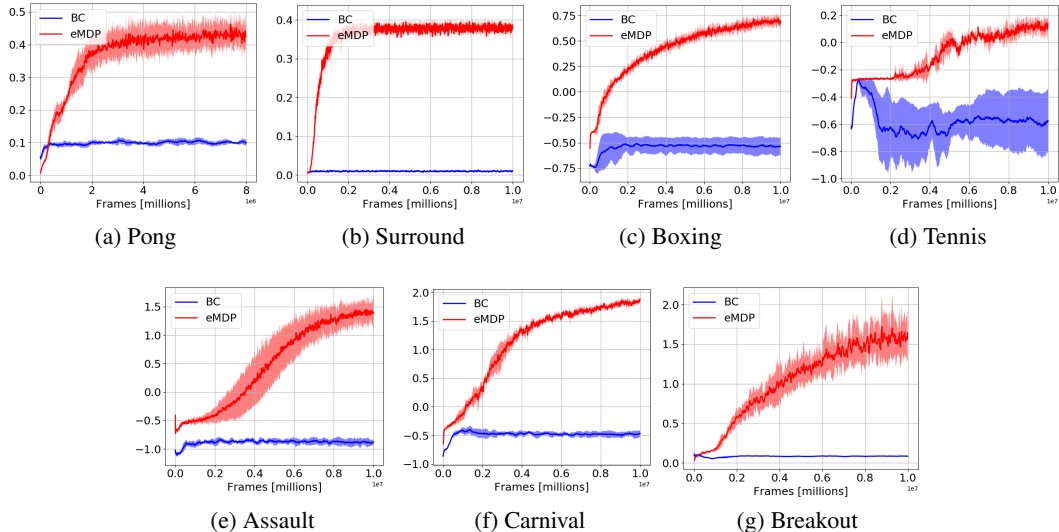

Figure 2: **Performance Comparison**: results on 7 games. The $x$-axis represent the number of game frames and the $y$-axis Numbers represent the cumulative return from the $e$MDP model.

On the other hand, the BC solution will seek to find the mode of the posterior distribution defined by the training set and the prior function. For simplicity, we assume a coherent training set (i.e., that $\forall s_i = s_j : a_i = a_j$), although a similar result can be drawn in the general case. We also assume a uniform prior function on the hypothesis set of $\tau$-regularized policies: $\Pi_\tau = \{\pi \,|\, \forall (a, s) \in \mathcal{A} \times \mathcal{S} : \pi(a|s) \geq \tau\}$. Lemma 6.2 shows that the optimal solution of BC assigns minimal probability to each sub-optimal action and a probability of $1 - (|\mathcal{A}| - 1)\tau$ to the ground-truth one.

**Lemma 6.2.** *Assume a set of policies $\Pi_\tau$ with a uniform prior function and a coherent training set. The solution of BC is given by:*

$$\pi^{bc}(a|s) = \begin{cases} 1 - (|\mathcal{A}| - 1)\tau, & \text{if } y(s) = a. \\ \tau, & \text{else} \end{cases} \tag{10}$$

*Proof.* See Appendix F. $\square$

We can see that the optimal solution of BC distributes the probability evenly across all non-optimal actions, regardless of what next state they induce. On the other hand, the optimal solution of $e$MDP assigns probabilities proportionally to their immediate reward which is directly related to the error in the state space which accounts for the deviation between the agent and the expert. Thus, while BC is oblivious to action outcomes, $e$MDP considers the dynamics of the problem.

## 7 CONCLUSIONS

We introduced the $e$MDP model as a formulation of solving imitation problems using RL, without requiring full knowledge about the state dynamics. $e$MDP can augment a training set by simulating the components of the state space for which the transition model is known. However, the degree to which the synthetic augmentation is reliable highly depends on the extensiveness of the responsive kernel. For example, when the responsive kernel is minimal ($S_r \to \emptyset, S_u \to S$), so is the level of synthetic augmentation. However, in this case the reward will be highly indicative of the task at hand because we fix a large portion of $\mathcal{S}$. Overall, we identify a tension between the reliability of the model (measured in the reward), and its level of augmentation (See Appendix N for a further discussion about kernel extensiveness). An additional factor that affects the $e$MDP model but was not accounted for in our analysis is the level of interaction between $\mathcal{F}_r$ and $\tilde{\mathcal{F}}_u$. As future research, it would be interesting to suggest an improved model that can quantify the level of interaction and use it to reward policies that follow the expert while leading to low $\mathcal{F}_r/\tilde{\mathcal{F}}_u$ interaction.

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

# Appendix

## A  OPTIMAL SOLUTION OF THE $e$MDP MODEL

The following Lemma ensures that the optimal solution of the $e$MDP model coincides with the optimal solution of Eq 2.

**Lemma A.1** ($e$MDP Optimal Solution). *Let $\mathcal{K}(p,q)$ be an IPM measure over probability distributions $p, q$. Let $d_\pi^t, \tilde{d}_\pi^t$ denote the $t$-step state distributions induced by $\tilde{\mathcal{F}}, \{\mathcal{F}_r, \mathcal{F}_u\}$ respectively. Then, if $\sum_{t=1}^{\infty} \gamma^t \mathcal{K}(d_\pi^t, d_E^t) = 0$ then $\tilde{d}_\pi^t = d_\pi^t$ for all $t \geq 0$*

*Proof.* Let $\mathcal{F}_r, \mathcal{F}_u$ be the responsive and unresponsive kernels in the original MDP. I.e., $s_r^{(t+1)} = \mathcal{F}_r(s_t, a_t), s_u^{(t+1)} = \mathcal{F}_u(s_t, a_t)$. Let $\tilde{\mathcal{F}}_r, \tilde{\mathcal{F}}_u$ be the responsive and unresponsive kernels in the $e$MDP model. I.e., $s_r^{(t+1)} = \mathcal{F}_r(s_t, a_t), s_u^{(t+1)} = tilde\mathcal{F}_u(s_t, a_t)$. The responsive kernels are equivalent by definition: $\tilde{\mathcal{F}} \equiv \mathcal{F}$. Regarding the unresponsive kernel, we write:

$$\tilde{\mathcal{F}}(s_t, t) = s_{u,e}^{(t+1)} = \mathcal{F}_u(s_t, \pi_E(s_t) = \mathcal{F}_u(s_t, a_t) \Big|_{a_t = \pi_E}.$$

Therefore, we get an equivalence in the unresponsive transition kernels at the point where actions are taken according to the expert.

Using the kernel equivalence along with the fact that the initial state distribution is the same, $\tilde{d}_\pi^0 = d_\pi^0$, it can be easily shown in induction that the state distribution remains the same for every step $t$. $\qquad\square$

## B  NORM FUNCTION INEQUALITY

**Lemma B.1** (Norm Function Inequality). *Let $f$ be a norm function. Then $\forall x, y$ the following holds:*

$$|f(x) - f(y)| \leq |f(x - y)|$$

*Proof.* Using Minkowski's inequality we have that:

$$f(x) + f(y - x) \geq f(x + y - x) = f(y)$$

Which leads to:

$$f(y - x) \geq f(y) - f(x)$$

And in the same way, since $f$ is symmetric we have that:

$$f(y) + f(x - y) \geq f(y + x - y) = f(x)$$

Which leads to:

$$f(x - y) = f(y - x) \geq f(x) - f(y)$$

Combining the above we get that $\forall x, y$:

$$|f(y) - f(x)| \leq f(y - x) = |f(y - x))|$$

$\qquad\square$

## C  RESPONSIVE REWARD

Consider a self-driving imitation task where a policy car $S_r$ follows an expert car $S_{r,e}$. $S_r$ is penalized via the Metric-reward for deviating from $S_{r,e}$: $r_{metric} = ||S_r - S_{r,e}||_{d_\mathcal{S}}$. However, prior knowledge tells us that $S_r$ should also maintain distance from other cars in the scene: $S_{u,e}$. Therefore, extra cost terms between $S_r$ and $S_{u,a}$ can be added.

## D    RESPONSIVE TERMINATION

Re-consider the self-driving example. We know that an accident or even a "bump" with other cars should terminate the current episode. Therefore, we can design an auxiliary responsive termination signal by applying prior knowledge on the relations between $S_r$ and $S_{u,e}$.

## E    THE OPTIMAL SOLUTION OF ENTROPY REGULARIZED RL AT THE BC REGIME

**Lemma E.1.** *The solution of optimization problem 5 at the bc-regime ($\gamma = 0$) is given by:*

$$\pi^{RL}(a|s) = \frac{e^{r(s,a)/\tau}}{\sum_{\tilde{a}} e^{r(s,\tilde{a})/\tau}} \tag{11}$$

*Proof.* It was previously shown by Nachum et al. (2017) that the causal entropy term can be written in the following recursive way:

$$\mathcal{H}(s,\pi) = \sum_a \pi(a|s)\big[ -\log \pi(a|s) + \gamma \mathcal{H}(s',\pi)\big],$$

which allows us to draw a similar recursive connection between the optimal value function $V^*(s)$ and the optimal policy:

$$\pi^{RL}(a|s) = \frac{e^{\big(r(s,a)+\gamma V^*(s')\big)/\tau}}{e^{V^*(s)/\tau}}.$$

We use the fact that $\gamma = 0$ at the BC regime to get that:

$$
\begin{aligned}
\frac{e^{\big(r(s,a)+\gamma V^*(s')\big)/\tau}}{e^{V^*(s)/\tau}} &= \frac{e^{\big(r(s,a)+\gamma V^*(s')\big)/\tau}}{e^{\log \sum_{\tilde{a}} e^{\big(r(s,\tilde{a})+\gamma V^*(s')\big)/\tau}}} \\
&= \frac{e^{\big(r(s,a)+\gamma V^*(s')\big)/\tau}}{\sum_{\tilde{a}} e^{\big(r(s,\tilde{a})+\gamma V^*(s')\big)/\tau}} = \frac{e^{r(s,a)/\tau}}{\sum_{\tilde{a}} e^{r(s,\tilde{a})/\tau}}
\end{aligned}
\tag{12}
$$

$\square$

## F    THE OPTIMAL SOLUTION OF ENTROPY REGULARIZED BEHAVIOR CLONING

**Lemma F.1.** *Assume a set of policies $\Pi_\tau$ with a uniform prior function and a coherent training set. The solution of behavior cloning is given by:*

$$\pi^{bc}(a|s) = \begin{cases} 1 - (|\mathcal{A}-1|)\tau, & \text{if } y(s) = a. \\ \tau, & \text{else} \end{cases} \tag{13}$$

*Proof.* Entropy-regularized behavior cloning aims to solve the following *maximum a posteriori* optimization problem:

$$\pi^{bc} = \arg\max_\pi P(\pi|\mathcal{D}). \tag{14}$$

Using Bayes rule and the *naive* assumption we have that:

$$
\begin{aligned}
\arg\max_\pi P(\pi|\mathcal{D}) &= \arg\max_\pi \frac{P(\mathcal{D}|\pi)P(\pi)}{P(\mathcal{D})} \\
&= \arg\max_\pi P(\pi)\Pi_{(s,a)\in\mathcal{D}}P(s,a|\pi) = \arg\max_\pi \log P(\pi) + \sum_{(s,a)\in\mathcal{D}} \log \pi(a|s)
\end{aligned}
\tag{15}
$$

Define $\Pi_\tau$ to be the set of policies with a $\tau$ lower bound on the allowed conditional probability:

$$\Pi_\tau = \{\pi \mid \forall (a, s) \in \mathcal{A} \times \mathcal{S} : \pi(a|s) \geq \tau\}.$$

The prior function assigns zero-probability to functions outside of $\Pi_\tau$, and an even probability measure of $\frac{1}{|\Pi_\tau|}$ to each of its members. Therefore, we can limit the search to policies that reside in $\Pi_\tau$:

$$\pi^{bc} = \arg\max_{\pi \in \Pi_\tau} \log P(\pi) + \sum_{(s,a) \in \mathcal{D}} \log \pi(a|s)$$

. Since the prior assigns an even distribution to all members in the set, we can remove the first term and get that:

$$\pi^{bc} = \arg\max_{\pi \in \Pi_\tau} \sum_{(s,a) \in \mathcal{D}} \log \pi(a|s)$$

.

Removing the $\log$ function we end up with:

$$\pi^{bc} = \arg\max_{\pi \in \Pi_\tau} \sum_{(s,a) \in \mathcal{D}} \pi(a|s)$$

From here it easy to see that the policy that maximizes the expression above is the one that assigns the maximum allowed probability to the ground-truth action while evenly distributing the remaining weight among the other ones:

$$\pi^{bc}(a|s) = \begin{cases} 1 - (|\mathcal{A} - 1|)\tau, & \text{if } y(s) = a. \\ \tau, & \text{else} \end{cases} \tag{16}$$

$\square$

## G  TRACTABLE COMPUTATION OF THE EMPIRICAL PROBABILITY DISTANCE ESTIMATOR

Calculating the estimator in ( 6) requires to solve an optimization problem. Therefore, it is not straightforward to calculate. In the following, we present an easier form of the estimator as a solution to a linear program.

**Lemma G.1.** *For all $\alpha \in [0, 1]$ the following function attains the supremum in 6:*

$$f_\alpha(s) := \alpha \min_{i=1,\ldots 2m} (a_i^* + d_{\mathcal{S}}(s, \tilde{S}_i)) \\ + (1 - \alpha) \max_{i=1,\ldots 2m} (a_i^* - d_{\mathcal{S}}(s, \tilde{S}_i)), \tag{17}$$

*where $\{a_i^*\}_{i=1}^{2m}$ solve the following linear program:*

$$\max_{a_1,\ldots a_{2m}} \frac{1}{m} \sum_{i=1}^{2m} c_i a_i \tag{18}$$

*such that:*

$$-d_{\mathcal{S}}(\tilde{S}_i, \tilde{S}_j) \leq a_i - a_j \leq d_{\mathcal{S}}(\tilde{S}_i, \tilde{S}_j), \forall i, j.$$

*Proof.* See Theoram 2.1 in Sriperumbudur et al. (2012)  $\square$

## H  CONVERGENCE RATE OF THE EMPIRICAL ESTIMATOR

The points we sample from the *e*MDP model are a function of the specific realization of expert demonstrations $\mathcal{D}$. Different demonstration sets will lead to different estimations. In the following, we draw a PAC result on the convergence rate of the empirical estimator.

**Lemma H.1.** *Let $\mathcal{F}$ be the space of measurable functions such that $||f||_\infty \leq \nu$, , $Var_{\{P,Q\}}(f) \leq \sigma^2_{\{P,Q\}} \; \forall f \in \mathcal{F}$. Let $C_0, C_1 < \infty$, $\alpha > 0$, $\delta \in (0,1)$, $D = \{\tilde{S}_i\}_{i=0}^{2m-1} \sim P \times \hat{P}$. Then $\forall s$ the following holds:*

$$P\Big(\mathcal{K}\big(P_{\mathcal{M}}(\cdot|s), P_{\hat{\mathcal{M}}}(\cdot|s)\big) >$$
$$\hat{\mathcal{K}}\big(\mathbb{P}_{\mathcal{M}}(\cdot|s), \mathbb{P}_{\hat{\mathcal{M}}}(\cdot|s)\big) + \epsilon(m,\delta,c,\nu,C_0,C_1)\Big) \leq 2e^{-c}$$

*where:*

$$
\begin{aligned}
\epsilon_{m,\delta,c,\nu,C_0,C_1} =\; & \frac{2(1+\alpha)}{1-\delta}\left(C_0\sqrt{\frac{2}{m}} + C_1 \sqrt[d+1]{\frac{1}{m}}\right) \\
& + \frac{1}{m}\sqrt{4c \max\{\sigma^2_{\{P,\hat{P}\}}\}} \\
& + \frac{4c\nu}{m}\left(\frac{2}{3} + \frac{1}{\alpha} + \frac{1+\alpha}{\delta(1-\delta)}\right)
\end{aligned}
\tag{19}
$$

*Proof.* See Theorem 3.3 in Sriperumbudur et al. (2012) ∎

## I  *e*MDP VALUE FUNCTION ERROR

In the following, we present the main result that relates the $n-$step divergence to the error in the value function. The following lemma is true for all $s$:

**Lemma I.1.** *Let $\mathcal{M}, \hat{\mathcal{M}}$ be two MDPs with Lipschitz transition functions with constants $L, \hat{L}$ respectively such that $\bar{L} = \min\{L, \hat{L}\}$, and Lipschitz reward functions with a constant $L_R$. Then, $\forall s$, $\bar{L} \in [0, \frac{1}{\gamma})]$ and $n \geq 1$, with probability at least $1 - 2e^{-c}$ the following holds:*

$$|V_{\mathcal{M}}(s) - V_{\hat{\mathcal{M}}}(s)| \leq \frac{\gamma L_R \bar{K}_{\mathbb{P}_{\mathcal{M},\hat{\mathcal{M}}}}}{(1-\gamma)(1-\gamma\bar{L})} \tag{20}$$

*Proof.* (Following Theorem 2 in Asadi et al. (2018)) Let $\delta_s$ be a Dirac delta function in $s$, then, for each state $s$, the difference between the value function under $\mathcal{M}$ and $\hat{\mathcal{M}}$ is given by:

$$V_{\mathcal{M}}(s) - V_{\hat{\mathcal{M}}}(s) = \sum_{n=0}^{\infty} \gamma^n \int r(s')\big(\mathcal{P}_{\mathcal{M}}^n(s'|\delta_s) - \mathcal{P}_{\hat{\mathcal{M}}}^n(s'|\delta_s)\big)ds'$$

Define the function $f(s) = \frac{r(s)}{L_R}$ where $L_R$ is the Lipschitz constant of $r(s)$. It can be easily shown that $||f||_L = 1$. With the definition of $f(s)$ we can re-write the state value gap as:

$$V_{\mathcal{M}}(s) - V_{\hat{\mathcal{M}}}(s) = L_R \sum_{n=0}^{\infty} \gamma^n \int f(s')\big(\mathcal{P}_{\mathcal{M}}^n(s'|\delta_s) - \mathcal{P}_{\hat{\mathcal{M}}}^n(s'|\delta_s)\big)ds'$$

Since $f(s) \in \mathcal{F}$ where $\mathcal{F} = \{f : ||f||_L \leq 1\}$ we can bound the state value gap from above by taking the supremum over $\mathcal{F}$:

$$
\begin{aligned}
V_{\mathcal{M}}(s) - V_{\hat{\mathcal{M}}}(s) =\; & L_R \sum_{n=0}^{\infty} \gamma^n \int f(s')\big(\mathcal{P}_{\mathcal{M}}^n(s'|\delta_s) - \mathcal{P}_{\hat{\mathcal{M}}}^n(s'|\delta_s)\big)ds' \\
\leq\; & L_R \sum_{n=0}^{\infty} \gamma^n \int_{\sup_{f \in \mathcal{F}}} f(s')\big(\mathcal{P}_{\mathcal{M}}^n(s'|\delta_s) - \mathcal{P}_{\hat{\mathcal{M}}}^n(s'|\delta_s)\big)ds'
\end{aligned}
$$

Following the definition of IPM in Eq equation 1, we see that the right hand side is exactly the Wasserstein distance between the state distribution induced after taking $n$ steps in $\mathcal{M}$ and $\hat{\mathcal{M}}$:

$$V_{\mathcal{M}}(s) - V_{\hat{\mathcal{M}}}(s) = L_R \sum_{n=0}^{\infty} \gamma^n \mathcal{K}\big(P_{\mathcal{M}}^n(\cdot|\delta_s), P_{\hat{\mathcal{M}}}^n(\cdot|\delta_s)\big)$$

Relying on Lemma 5.1, with probability of at least $1 - 2e^{-c}$ we can upper bound the $n$-step deviation as follows:

$$L_R \sum_{n=0}^{\infty} \gamma^n \mathcal{K}\big(P_{\mathcal{M}}^n(\cdot|\delta_s), P_{\hat{\mathcal{M}}}^n(\cdot|\delta_s)\big) \leq L_R \sum_{n=0}^{\infty} \gamma^n \bar{K}_{\mathbb{P}_{\mathcal{M},\hat{\mathcal{M}}}} \sum_{i=0}^{n-1} \bar{L}^i.$$

Re-writing the right hand side we get that:

$$L_R \sum_{n=0}^{\infty} \gamma^n \bar{K}_{\mathbb{P}_{\mathcal{M},\hat{\mathcal{M}}}} \sum_{i=0}^{n-1} \bar{L}^i = L_R \bar{K}_{\mathbb{P}_{\mathcal{M},\hat{\mathcal{M}}}} \sum_{n=0}^{\infty} \gamma^n \frac{1 - \bar{L}^n}{1 - \bar{L}}$$

And after re-arranging we can write that:

$$L_R \bar{K}_{\mathbb{P}_{\mathcal{M},\hat{\mathcal{M}}}} \sum_{n=0}^{\infty} \gamma^n \frac{1 - \bar{L}^n}{1 - \bar{L}} = L_R \frac{\bar{K}_{\mathbb{P}_{\mathcal{M},\hat{\mathcal{M}}}}}{1 - \bar{L}} \Big[ \sum_{n=0}^{\infty} \gamma^n - \sum_{n=0}^{\infty} (\gamma \bar{L})^n \Big]$$

Using on the fact that $\bar{L} \in [0, \frac{1}{\gamma})]$ we know that the right hand side is finite:

$$L_R \frac{\bar{K}_{\mathbb{P}_{\mathcal{M},\hat{\mathcal{M}}}}}{1 - \bar{L}} \Big[ \sum_{n=0}^{\infty} \gamma^n - \sum_{n=0}^{\infty} (\gamma \bar{L})^n \Big] = L_R \frac{\bar{K}_{\mathbb{P}_{\mathcal{M},\hat{\mathcal{M}}}}}{1 - \bar{L}} \Big[ \frac{1}{1 - \gamma} - \frac{1}{1 - \gamma \bar{L}} \Big]$$

Which can be written as:

$$L_R \frac{\bar{K}_{\mathbb{P}_{\mathcal{M},\hat{\mathcal{M}}}}}{1 - \bar{L}} \Big[ \frac{1}{1 - \gamma} - \frac{1}{1 - \gamma \bar{L}} \Big] == L_R \frac{\bar{K}_{\mathbb{P}_{\mathcal{M},\hat{\mathcal{M}}}}}{1 - \bar{L}} \Big[ \frac{\gamma(1 - \bar{L})}{(1 - \gamma)(1 - \gamma \hat{L})} \Big] = \frac{\gamma L_R \bar{K}_{\mathbb{P}_{\mathcal{M},\hat{\mathcal{M}}}}}{(1 - \gamma)(1 - \gamma \hat{L})}$$

Since $\mathcal{K}$ equipped with $\mathcal{F} = \{f : ||f||_L \leq 1\}$ leads to the Wasserstein distance which is a metric, it is therefore symmetric. We can repeat the process, this time with $V_{\mathcal{M}}(s) - V_{\hat{\mathcal{M}}}(s)$, which concludes the proof. $\qquad\square$

## J    COMPOSITION LEMMAS

We wish to make our result general to several IPMs besides the Wasserstein metric. In the following we present two helpful lemmas.

### J.1    LIPSCHITZ COMPOSITION

The following lemma (Following Lemma 2 in Asadi et al. (2018)) shows that the composition of two Lipschitz functions is also Lipschitz constant:

**Lemma J.1.** *Define 3 metric spaces* $(M_1, d_1), (M_2, d_2), (M_3, d_3)$. *Define Lipschitz functions* $g : M_1 \to M_2$, $f : M_2 \to M_3$, *then* $h : f \circ g : M_1 \to M_3$ *is also Lipschitz with a constant bounded above by* $||g||_L ||f||_L$

*Proof.*

$$\begin{aligned}
||h||_L &= \sup_{s_1, s_2 \in M_1} \frac{d_3(f(g(s_1)), f(g(s_2)))}{d_1(s_1, s_2)} \\
&= \sup_{s_1, s_2 \in M_1} \frac{d_2(g(s_1), g(s_2))}{d_1(s_1, s_2)} \frac{d_3(f(g(s_1)), f(g(s_2)))}{d_2(g(s_1), g(s_2))} \\
&\leq \sup_{s_1, s_2 \in M_1} \frac{d_2(g(s_1), g(s_2))}{d_1(s_1, s_2)} \sup_{s_1', s_2' \in M_2} \frac{d_3(f(g(s_1')), f(g(s_2')))}{d_2(g(s_1'), g(s_2'))} = ||g||_L ||f||_L
\end{aligned} \tag{21}$$

$\qquad\square$

## J.2 DUDLEY COMPOSITION

Recall that the Dudley metric can be extracted from the IPM formulation (Eq. 1) by setting

$$\mathcal{F} = \{f : ||f||_{BL} \leq 1\},$$

where $||f||_{BL} := ||f||_\infty + ||f||_L$, and $||f||_\infty := \sup\{|f(x)| : x \in \mathcal{S}\}$. The following lemma states that the composition of two function with a bounded $BL$ norm has a bounded $BL$ norm itself.

**Lemma J.2.** *Define 3 metric spaces* $(M_1, d_1), (M_2, d_2), (M_3, d_3)$. *Define functions* $g : M_1 \to M_2$, $f : M_2 \to M_3$. *Then* $h : f \circ g : M_1 \to M_3$ *has a bounded* $BL$ *norm:* $||h||_{BL} \leq ||g||_L ||f||_L + ||f||_\infty$

*Proof.*

$$
\begin{aligned}
||h||_{BL} &= \sup_{s_1, s_2 \in M_1} \frac{d_3(f(g(s_1)), f(g(s_2)))}{d_1(s_1, s_2)} + \sup_{s_3 \in M_1} |f(g(s_3))| \\
&\leq ||g||_L ||f||_L + \sup_{s' \in M_1} |f(s')| \\
&= ||g||_L ||f||_L + ||f||_\infty
\end{aligned}
\tag{22}
$$

where the inequality is given by Lemma J.1. □

## J.3 TOTAL VARIATION COMPOSITION

Recall that the TV metric can be extracted from the IPM formulation (Eq. 1) by setting

$$\mathcal{F} = \{f : ||f||_\infty \leq 1\},$$

where $||f||_\infty := \sup\{|f(x)| : x \in \mathcal{S}\}$. The following lemma states that the composition of two functions with a bounded TV norm has a bounded TV norm itself:

**Lemma J.3.** *Define 3 metric spaces* $(M_1, d_1), (M_2, d_2), (M_3, d_3)$. *Define functions* $g : M_1 \to M_2$, $f : M_2 \to M_3$. *Then* $h : f \circ g : M_1 \to M_3$ *has a bounded TV norm:* $||h||_\infty \leq ||f||_\infty$

*Proof.*

$$||h||_\infty = \sup_{s \in M_1} |f(g(s))| \leq \sup_{s' \in M_2} |f(s')| = ||f||_\infty \tag{23}$$

□

## K FROM 1-STEP DIVERGENCE TO $n-$STEP DIVERGENCE

We prove this lemma following the proof Theorem 1 in Asadi et al. (2018). We generalize the result to several IPM measures using the composition lemmas we proved in Appendix J.

**Lemma K.1.** *Let* $\mathcal{M}, \hat{\mathcal{M}}$ *be two MDPs with Lipschitz transition functions with constants* $L, \hat{L}$ *respectively. Let* $\bar{L} = \min\{L, \hat{L}\}$, *and denote* $\bar{K}_{\mathbb{P}_{\mathcal{M}, \hat{\mathcal{M}}}} = \max_{s \in \mathcal{D}} \hat{\mathcal{K}}\big(\mathbb{P}_{\mathcal{M}}(\cdot|s), \mathbb{P}_{\hat{\mathcal{M}}}(\cdot|s)\big) + \epsilon_{m, \delta, c, \nu, C_0, C_1}$, *then for all* $n \geq 1$ *the following holds:*

$$P\Big(\mathcal{K}\big(P_{\mathcal{M}}^n(\cdot|\mu), P_{\hat{\mathcal{M}}}^n(\cdot|\mu)\big) \geq \bar{K}_{\mathbb{P}_{\mathcal{M}, \hat{\mathcal{M}}}} \sum_{i=0}^{n-1} \bar{L}^i \Big) \leq 2e^{-c}, \tag{24}$$

*where* $P_{\mathcal{G}}^n(\cdot|\mu)$ *is the state distribution induced by model* $\mathcal{G}$ *after* $n$ *steps taken from an initial distribution* $\mu$.

*Proof.* The proof is done by induction.
**Induction base**: Using the definition of an IPM we can write that:

$$\mathcal{K}(P_{\mathcal{M}}(\cdot|\mu), P_{\hat{\mathcal{M}}}(\cdot|\mu)) = \sup_{f \in \mathcal{F}} \int \int \big(P_{\mathcal{M}}(s'|s) - P_{\hat{\mathcal{M}}}(s'|s)\big) f(s')\mu(s)dsds'$$

$$\leq \int \sup_{f \in \mathcal{F}} \int \big(P_{\mathcal{M}}(s'|s) - P_{\hat{\mathcal{M}}}(s'|s)\big) f(s')ds'\mu(s)ds = \int \mathcal{K}\big(P_{\mathcal{M}}(s'|s), P_{\hat{\mathcal{M}}}(s'|s)\big)\mu(s)ds.$$

We can use the fact that $\int_{\mathcal{S}} \mu(s)ds = 1$ to get:

$$= \mathcal{K}\big(P_{\mathcal{M}}(s'|s), P_{\hat{\mathcal{M}}}(s'|s)\big) \int \mu(s)ds = \mathcal{K}\big(P_{\mathcal{M}}(s'|s), P_{\hat{\mathcal{M}}}(s'|s)\big).$$

With probability at least $1 - 2e^{-c}$ on the choice of $\mathcal{D}$, the above can be bounded above by $\bar{K}_{\mathbb{P}_{\mathcal{M},\hat{\mathcal{M}}}}$ which concludes the proof of the base of the induction.

**Induction step**: Assume that the claim holds for $n - 1$, i.e.:

$$\mathcal{K}\big(P_{\mathcal{M}}^{n-1}(\cdot|\mu), P_{\hat{\mathcal{M}}}^{n-1}(\cdot|\mu)\big) \leq \bar{K}_{\mathbb{P}_{\mathcal{M},\hat{\mathcal{M}}}} \sum_{i=0}^{n-2} \bar{L}^i$$

Using the triangle inequality we write that:

$$\mathcal{K}\big(P_{\mathcal{M}}^n(\cdot|\mu), P_{\hat{\mathcal{M}}}^n(\cdot|\mu)\big) \leq \mathcal{K}\big(P_{\hat{\mathcal{M}}}^n(\cdot|\mu), P_{\hat{\mathcal{M}}}(\cdot|P_{\mathcal{M}}^{n-1}(\cdot|\mu))\big) + \mathcal{K}\big(P_{\hat{\mathcal{M}}}(\cdot|P_{\mathcal{M}}^{n-1}(\cdot|\mu)), P_{\mathcal{M}}^n(\cdot|\mu)\big)$$

$$= \mathcal{K}\big(P_{\hat{\mathcal{M}}}(\cdot|P_{\hat{\mathcal{M}}}^{n-1}(\cdot|\mu)), P_{\hat{\mathcal{M}}}(\cdot|P_{\mathcal{M}}^{n-1}(\cdot|\mu))\big) + \mathcal{K}\big(P_{\hat{\mathcal{M}}}(\cdot|P_{\mathcal{M}}^{n-1}(\cdot|\mu)), P_{\mathcal{M}}(\cdot|P_{\mathcal{M}}^{n-1}(\cdot|\mu))\big).$$

$$\leq \mathcal{K}\big(P_{\hat{\mathcal{M}}}(\cdot|P_{\hat{\mathcal{M}}}^{n-1}(\cdot|\mu)), P_{\hat{\mathcal{M}}}(\cdot|P_{\mathcal{M}}^{n-1}(\cdot|\mu))\big) + \bar{K}_{\mathbb{P}_{\mathcal{M},\hat{\mathcal{M}}}}.$$

To upper bound the first term we rely on the proof of the Kantarovich-Rubinstein duality theorem (See Asadi et al. (2018), page 3 13). It is easy to see that using our composition lemmas (See Appendix J), the result proved there can be generalized in several ways: 1) A Dudley-metric composed on a transition model with a bounded $BL$ norm. 2) a TV-metric composed on a transition model with a bounded $TV$ norm. Therefore:

$$\mathcal{K}\big(P_{\hat{\mathcal{M}}}(\cdot|P_{\hat{\mathcal{M}}}^{n-1}(\cdot|\mu)), P_{\hat{\mathcal{M}}}(\cdot|P_{\mathcal{M}}^{n-1}(\cdot|\mu))\big) \leq ||P|| \mathcal{K}\big(P_{\hat{\mathcal{M}}}^{n-1}(\cdot|\mu)), P_{\mathcal{M}}^{n-1}(\cdot|\mu)\big)$$

where $||P_{\hat{\mathcal{M}}}||$ is the transition-model norm. E.g., $\hat{L}$ when reducing IPM to Wasserstein-metric together with a Lipschitz transition model as we shell assume for the rest of the proof:

$$\mathcal{K}\big(P_{\hat{\mathcal{M}}}(\cdot|P_{\hat{\mathcal{M}}}^{n-1}(\cdot|\mu)), P_{\hat{\mathcal{M}}}(\cdot|P_{\mathcal{M}}^{n-1}(\cdot|\mu))\big) \leq \hat{L}\mathcal{K}\big(P_{\hat{\mathcal{M}}}^{n-1}(\cdot|\mu)), P_{\mathcal{M}}^{n-1}(\cdot|\mu)\big)$$

Finally:

$$\mathcal{K}\big(P_{\mathcal{M}}^n(\cdot|\mu), P_{\hat{\mathcal{M}}}^n(\cdot|\mu)\big) \leq \hat{L}\mathcal{K}\big(P_{\hat{\mathcal{M}}}^{n-1}(\cdot|\mu)), P_{\mathcal{M}}^{n-1}(\cdot|\mu)\big) + \bar{K}_{\mathbb{P}_{\mathcal{M},\hat{\mathcal{M}}}} \leq \bar{K}_{\mathbb{P}_{\mathcal{M},\hat{\mathcal{M}}}} \sum_{i=0}^{n-1} \bar{L}^i$$

$\square$

## L    POLICY NETWORK ARCHITECTURE & TECHNICAL DETAILS

The CNN architecture used by all agents is described in Table 1. The state is a concatenation of last four frames, where each frame is $80 \times 80$ gray-level pixels. Although our method is agnostic to the action space, all environments we tested have discrete action spaces (not intentional). We used the Adam optimizer Kingma & Ba (2014) across all experiments. We used a linearly decaying learning rate with an initial coefficient of $7e-4$. All agents were trained on a single workstation with a single GPU/CPU.

| Layer Type | Size | Details | Activation |
|---|---|---|---|
| 2D Convolution | 32 filters | kernel size = 8, stride=4 | Relu |
| 2D Convolution | 64 filters | kernel size = 4, stride=2 | Relu |
| 2D Convolution | 64 filters | kernel size = 2, stride=1 | Relu |
| Fully Connected | 512 | | Relu |
| Fully Connected | $|\mathcal{A}|$ | | |

Table 1: **Policy Network Architecture**

## M   FURTHER EXPERIMENTAL RESULTS

Videos results can be found in the following link: `https://drive.google.com/drive/folders/1FSveO1JLNn4bNoP9FvfKHLRYnSYifxCn`. Note that the expert is overlaid in red for visualization purposes only.

## N   ON THE EFFECTS OF KERNEL EXTENSIVENESS

In the following, we discuss the effects of using an extensive kernel in the $e$MDP model, in the case of extensive knowledge about the transition kernel, vs the case of using a minimal kernel in the case of minimal prior knowledge.

**BC Limit:** $S_r \to \emptyset, S_u \to S$ With insufficient prior-knowledge about the world, the responsive kernel is minimal. In this case, partial simulation barely enriches the demonstration set $\mathcal{D}$ with new simulated states. However, in this case, the reward signal provided by a PS-Env is highly indicative of the task at hand. This is true because we fix a large portion of the state space ($S_u \to S$). Therefore, the hypothetical behavior of the expert in the observed state $S$ should align to a high degree with the recorded behavior of the expert $S_{r,e}$ upon which we define $r_{metric}$.

**Model-Based Limit:** $S_r \to S, S_u \to \emptyset$ On the contrary, with extensive knowledge about the world, a comprehensive responsive kernel can be designed. Partial simulation can enrich the training set with numerous new simulated states. Yet, the reward signal provided at this limit is dubious: we fix a small subset of the state ($S_u \to \emptyset$), and as a result we have fewer guarantees on the hypothetical behavior of the expert in the observed state $S$ with respect to the recorded behavior $S_{r,e}$ upon which we reward the agent. To exemplify this problem, consider the extreme case where $\mathcal{S}_r \equiv \mathcal{S}$. At the presence of any randomness, the simulated trajectory will completely diverge from the reference trajectory, thus ruling out the utility of $r_{metric}$. The only case in which $r_{metric}$ maintains relevancy is in a completely deterministic environment.

