# OpenReview forum: "Partial Simulation for Imitation Learning"
_ICLR.cc/2020/Conference — Reject_

### Official Review · AnonReviewer3 · 2019-10-18
**Official Blind Review #3**

**Rating:** 3

**Review:**

* Paper summary.
The paper considers an IL problem where partial knowledge about the transition probability of the MDP is available. To use of this knowledge, the paper proposes an expert induced MDP (eMDP) model where the unknown part of transition probability is modeled as-is from demonstrations. Based on eMDP, the paper proposes a reward function based on an integral probability metric between a state distribution of expert demonstrations under the target MDP and a state distribution of the agent under the eMDP. Using this reward function, the paper proposes an IL method that maximizes this reward function by RL. The main theoretical result of the paper is that the error between value function of the target MDP and that of the eMDP can be upper-bounded. Empirical comparisons against behavior cloning on discrete control tasks show that the proposed method performs better.

* Rating.
The main contribution of the paper is the eMDP model, which enables utilizing prior knowledge about the target MDP for IL. While this idea is interesting, eMDP is too restrictive and its practical usefulness is unclear. Moreover, there are other issues that should be addressed such as clarity and experiments. I vote for weak rejection.

* Major comments:
- Limited practicality due to an assumption on the unresponsive transition kernel.
In eMDP, the unresponsive transition kernel is modeled from demonstrations by directive using the observed next states (in demonstrations) as a next state of the agent. This modeling implicitly assumes that the agent cannot influence the unresponsive part of the state space. This is too restrictive, since actions of the agent usually influence other parts of the state space such as opponents and objects. For instance, in pong, actions of the agent indeed influence trajectories of the ball. Due to this restrictive assumption, I think the practicality of eMDP is too limited.

Also, it is unclear what happens when the transition of unresponsive state space is stochastic. In this case, the unresponsive transitions of eMDP would be incorrect since eMDP assumes deterministic transitions as-is from demonstrations. Though I am not sure about this.

- The equality in Eq. (3) should be an upper-bound. The IPM is defined by the absolute difference between summations (expectations), but the right-hand side of Eq. (3) is summations (expectations) of the absolute difference. These two are not equal, and the right-hand side should be an upper-bound.

- The paper is difficult to follow. There are many skipping contents in the paper. Especially, in the description of eMDP and its solution, where the paper describes the optimal solution of eMDP (Section 2) before describing the eMDP model (Section 4). Also, it is unclear from my first reading pass what is the actual IL procedure. Including a pseudo-code would help.

- The state-value function’s error bound ignores a policy. The proof utilizes the state distribution P(s’|s). However, this distribution should depend on a policy. It is unclear to me what is the policy function used in this error bound. Does this error-bound hold for any policy or only for the optimal policy? In the case that it only holds for the optimal policy, this result still does not provide useful guarantees when the optimal policy is not learned, similarly to the result in Proposition 2.1.

- Empirical evaluation lacks strong baseline methods. The paper compares the proposed method against behavior cloning, which is known to perform poorly under limited data. The paper requires stronger baseline methods such as those mentioned in Section 3. Also, it is strange that the vertical axis in Figure 2 represents the reward function of eMDP, which is an artificial quantity in Eq. (5). The results should present the reward function of the target MDP, which is the ground-truth reward of the task.

* Besides the major comments, I have few minor comments and questions:
- Does the reward function in Eq. (5) correspond to the Wasserstein distance for any metric d_s? If it is not, then the value function’s error-bound for the Wasserstein distance is not applicable to this reward function.
- What is the reward function r(s) in the value function’s error-bound that we can control the Lipschitz constant? We do not know the ground-truth reward r(s,a) so it cannot be controlled. The reward r(s,s’) of eMDP is given by the metric d_s of the state space so again we cannot control it.
- What is the metric d_s used in the experiments?
- How do you perform BC without expert actions in the experiments?
- Confusing notations. E.g., the function r is used for r(s,a) (MDP’s reward), r(s, s’) (eMDP’s reward), and r(s) (reward in the error bound’s proof); these r’s have different meaning and should be denoted differently.
- Typos. E.g., “Lemma 2.1” in Section 5 should be “Proposition 2.1”, “Lemma A.1” should be “Proposition 2.1”. "tildeF". etc.

*** After authors' response.
I read the other reviews and the response. My major concern was the impracticality of eMDP. While I can see that the eMDP model provides advantages over BC, I still think that it is impractical since it assumes the agent cannot influence the unresponsive part. Moreover, extracting next-unresponsive states as-is from demonstrations is problematic with stochastic transitions, since actual next-unresponsive states can be different from next-unresponsive states in demonstrations even when the sames state and actions are provided. Given that the major concern remains, I still vote for weak rejection.


**Experience Assessment:**

I have published one or two papers in this area.

**Review Assessment: Checking Correctness Of Derivations And Theory:**

I assessed the sensibility of the derivations and theory.

**Review Assessment: Checking Correctness Of Experiments:**

I assessed the sensibility of the experiments.

**Review Assessment: Thoroughness In Paper Reading:**

I read the paper at least twice and used my best judgement in assessing the paper.

---

> ### Author Response · Authors · 2019-11-08
> **response to review #3**
>
> We thank the reviewer for his detailed response. Below is our response to the main issues raised. Other issues will be fixed or handled in the PDF.
>
> •	Q: “…This modeling implicitly assumes that the agent cannot influence the unresponsive part of the state space. This is too restrictive…”
> •	A: The eMDP model bridges between BC and contemporary imitation methods by using the BC kernel for features we cant simulate, and the responsive kernel for features we can. While with BC the agent cant influence any part of the state space, our model allows some interaction to occur. Our experiments show that even partial interaction is better than zero interaction.
>
> •	Q: I think the practicality of eMDP is too limited.
> •	A: To the best of our knowledge, this is the first and only method that allows imitation in a multiplayer setup.
>
> •	Q: Also, it is unclear what happens when the transition of unresponsive state space is stochastic. In this case, the unresponsive transitions of eMDP would be incorrect since eMDP assumes deterministic transitions as-is from demonstrations. Though I am not sure about this.
> •	A: eMDP does not assume deterministic transitions.
>
> •	Q: The state-value function’s error bound ignores a policy. The proof utilizes the state distribution P(s’|s). However, this distribution should depend on a policy. It is unclear to me what is the policy function used in this error bound. Does this error-bound hold for any policy or only for the optimal policy?
> •	A: The value function error is true for any policy and not just for the optimal one, thus it was omitted. A comment about it should be added to the text.
>
> •	Q: Empirical evaluation lacks strong baseline methods. The paper compares the proposed method against behavior cloning, which is known to perform poorly under limited data. The paper requires stronger baseline methods such as those mentioned in Section 3.
> •	A: Unfortunately, this is not possible. Stronger baselines require full knowledge of F, which does not apply in our setting. For example, we cannot provide rules for the behavior of the opponent human player.
>
>
> •	Q: Does the reward function in Eq. (5) correspond to the Wasserstein distance for any metric d_s? If it is not, then the value function’s error-bound for the Wasserstein distance is not applicable to this reward function.
> •	A: The reward is defined over the metric space S while the Wasserstein distance is defined between probability distributions. In Section 5, we relate the metric distance to the distribution distance. Please see footnote 2 at the bottom of page 6.
>
> •	Q: What is the reward function r(s) in the value function’s error-bound that we can control the Lipschitz constant? We do not know the ground-truth reward r(s,a) so it cannot be controlled. The reward r(s,s’) of eMDP is given by the metric d_s of the state space so again we cannot control it.
> •	A: Yes. This is the reward from Eq. 5. It is determined by the state space and cannot be controlled.
>
> •	Q: What is the metric d_s used in the experiments?
> •	A: This is the reward from Eq. 5. We assume no knowledge about the state space and use a Dirac delta function at the event s_r=s_{r,e}. In some of the environments (that involve a shooting action) we implemented the Dirac function asymmetrically (agent shoots and experts don’t is rewarded differently than expert shoots and agents don’t).
>
> •	Q: How do you perform BC without expert actions in the experiments?
> •	A: The BC baseline “cheats” by looking at expert actions, while the eMDP model doesn't access A.

---

### Official Review · AnonReviewer2 · 2019-10-23
**Official Blind Review #2**

**Rating:** 6

**Review:**

This paper provides a theoretical basis called expert-induced MDP (eMDP) for formulating imitation learning through RL when only partial information on the transition kernel is given. Examples include imitating one player in multiplayer games where the learner cannot explicitly have the transition kernel for other players. eMDP just uses the available part of transition kernel (e.g. the target player's state-action transitions) for learning, but substitute the next state from demonstrations 'as is' into the unavailable part. In this situation, using IPM (Wasserstein-like) divergence, it is easily guaranteed that eMDP is consistent with the optimal solution in the limit. However, this does not guarantee anything when the optimum is not reached. The paper's main contributions would be to provides error bounds (Lemma 5.2 in particular) for the convergence rate in terms of Lipschitz constants and the maximum error between eMDP and the demonstration set. This roughly means that when transition functions are continuous and smooth and the sample demonstration set well characterize the underlying MDPs, then eMDP is theoretically guaranteed to work well. If these factors (Lipschitz continuity and the max deviation of state distributions conditioned on the given demonstration set) are expected to be small, eMDP can nicely perform learning with partial knowledge. Some demonstrations confirm eMDP works when simple BC fails in multiplayer games.

Overall, the paper is worth accepting. The proposed eMDP would be a technically simple solution, but providing a solid theoretical basis for that solution to these commonly seen situations of imitation learning with partial knowledge would be quite beneficial. The paper is well written with intuitive examples and nice overview summaries for the theoretically complicated parts. Though the implications of the experimental part are not clear, the paper fairly states this point and also provides a video for visual assessment with displaying ghost imitations.

One minor question: For the experimental part, BC baselines for multiplayer games just imitates the target player's behavior (player1), completely ignoring the states/actions of player2 (ignoring the states of bricks and balls for breakout-v0)?

**Experience Assessment:**

I do not know much about this area.

**Review Assessment: Checking Correctness Of Derivations And Theory:**

I assessed the sensibility of the derivations and theory.

**Review Assessment: Checking Correctness Of Experiments:**

I assessed the sensibility of the experiments.

**Review Assessment: Thoroughness In Paper Reading:**

I read the paper at least twice and used my best judgement in assessing the paper.

---

> ### Author Response · Authors · 2019-11-08
> **Response to review #2**
>
> We thank the reviewer for his positive feedback. Below is our response to the issues raised.
>
> •	Q: For the experimental part, BC baselines for multiplayer games just imitates the target player's behavior (player1), completely ignoring the states/actions of player2 (ignoring the states of bricks and balls for breakout-v0)?
> •	A: The BC baseline receives the original (un-factored) state as an input. In Breakout it is the complete frame including the bricks, ball and expert racket. The baseline is also granted access to expert action which we do not provide to the eMDP model. I.e., the BC baseline “cheats”.

---

### Official Review · AnonReviewer1 · 2019-10-23
**Official Blind Review #1**

**Rating:** 1

**Review:**

The paper proposes an imitation learning algorithm that learns from state-only demonstrations and assumes partial knowledge of the transition dynamics. The demonstrated states are decomposed into "responsive" and "unresponsive" features. The imitation policy is trained in an environment where the responsive state features are simulated and controllable by the agent, and the unresponsive state features are replayed from the demonstrations. The agent is rewarded for tracking the responsive state features in the demonstrations.

Overall, I was confused by this paper. Isn't it problematic that the imitation agent is trained in an environment where unresponsive state features from the demonstrations are replayed and not affected by the agent's actions? It would be nice to expand Section 4 to explain why it makes sense to reflect the unresponsive component in the transition kernel. It would also be helpful to include some of this information in Section 1.

I am also confused by the method. What is the purpose of observing unresponsive features, if the reward function for the imitation agent is only defined in terms of the responsive features? Is it that the imitation policy is conditioned on the unresponsive features?

There are several important experimental details missing from Sections 4 and 6. What distance metric was used to define the reward function? For example, was it Euclidean distance in pixel space for the Atari games? The main experimental results in Figure 2 are difficult to interpret without knowing the how the reward function is defined in the eMDP. Could the authors either provide definitions of the reward functions of the eMDPs in the experiments, or measure performance of the imitation agents using more interpretable or standardized metrics (e.g., game score) for the chosen environments?

It is also concerning that the Google Drive links to code and supplementary materials aren't anonymized.

**Experience Assessment:**

I have read many papers in this area.

**Review Assessment: Checking Correctness Of Derivations And Theory:**

I did not assess the derivations or theory.

**Review Assessment: Checking Correctness Of Experiments:**

I assessed the sensibility of the experiments.

**Review Assessment: Thoroughness In Paper Reading:**

I read the paper at least twice and used my best judgement in assessing the paper.

---

> ### Author Response · Authors · 2019-11-08
> **Response to Review #1**
>
> We thank the reviewer for his important comments and apologies for the issue with the link anonymization. Below is our response to the issues raised.
>
> •	Q: Isn't it problematic that the imitation agent is trained in an environment where unresponsive state features from the demonstrations are replayed and not affected by the agent's actions?
> •	A: Yes. Ideally, you would like to use a “full” simulation to eliminate the problem. However, most real-world problems do not have a simulator yet. Our experiments show that even partial interaction of the agent with the environment is better than zero interaction that happens with simulation free methods such as BC. Moreover, our transition kernel becomes less biased as the agent improves.
>
> •	Q: It would be nice to expand Section 4 to explain why it makes sense to reflect the unresponsive component in the transition kernel.
> •	A: Yes, we agree. The reflection operator is the transition kernel of BC where the prediction of the agent on a state does not affect the next state. The eMDP model bridges between BC and contemporary imitation methods by using the BC kernel for features we cant simulate, and the responsive kernel for features we can simulate. As stated before, the reflection operator also possesses the nice property of being unbiased when pi=pi_E and it requires zero calculations, thus dramatically accelerating training time.
>
> •	Q: What is the purpose of observing unresponsive features, if the reward function for the imitation agent is only defined in terms of the responsive features?
> •	A: The reward is defined over a) s_r: the responsive features of the agent which are observed in the state space, and b) s_{r,e}: the responsive features of the expert which are unobserved. How can the agent maximize the reward if it depends on unobserved features? It does so by observing s_u: the unresponsive features, which define (or at least have high mutual information with) s_{r,e}.
>
> •	Q: Is it that the imitation policy is conditioned on the unresponsive features?
> •	A: Yes, of course, the unresponsive features are a part of the state space and the agent and expert share the same state space. Think of Pong for example where the unresponsive features represent the opponent racket and the ball. The agent can't take actions without having this info.
>
> •	Q: What distance metric was used to define the reward function?
> •	A: We report the reward defined in Eq.5 (the optimization problem we wish to solve). In words, the reward is a Dirac delta function on the event s_r=s_{r,e}. It is the most restrictive form of similarity and requires zero prior knowledge. Calculating it requires extracting s_r and s_{r,e} at each step as explained in section 4. In the Atari games, we do the extraction manually from frame pixels. However, with access to the game’s ROM, this can be done much easier.

---

### Decision · Program_Chairs · 2019-12-19

**Decision:**

Reject

**Comment:**

The paper introduces the concept of an Expert Induced MDP (eMDP) to address imitation learning settings where environment dynamics are part known / part unknown. Based on the formulation a model-based imitation learning approach is derived and the authors obtain theoretical guarantees. Empirical validation focuses on comparison to behavior cloning. Reviewers raised concerns about the size of the contribution. For example, it is unclear to what degree the assumptions made here would hold in practical settings.